RESEARCH

# Chromatin spatial organization of wild type and mutant peanuts reveals high-resolution genomic architecture and interaction alterations

Xingguo Zhang[1†], Manish K. Pandey[2†], Jianping Wang[3†], Kunkun Zhao[1], Xingli Ma[1], Zhongfeng Li[1], Kai Zhao[1], Fangping Gong[1], Baozhu Guo[4*], Rajeev K. Varshney[2,5*] and Dongmei Yin[1*]

* Correspondence: yindm@henau.
edu.cn; r.k.varshney@cgiar.org;
rajeev.varshney@murdoch.edu.au;
baozhu.guo@usda.gov
†Xingguo Zhang, Manish K. Pandey
and Jianping Wang contributed
equally to this work.
[4]Crop Protection and Management
Research Unit, USDA-ARS, Tifton,
USA
[2]International Crops Research
Institute for the Semi-Arid Tropics
(ICRISAT), Hyderabad, India
[1]College of Agronomy, Henan
Agricultural University, Zhengzhou,
China
Full list of author information is
available at the end of the article

## Abstract

**Background:** Three-dimensional (3D) chromatin organization provides a critical foundation to investigate gene expression regulation and cellular homeostasis.

**Results:** Here, we present the first 3D genome architecture maps in wild type and mutant allotetraploid peanut lines, which illustrate A/B compartments, topologically associated domains (TADs), and widespread chromatin interactions. Most peanut chromosomal arms (52.3%) have active regions (A compartments) with relatively high gene density and high transcriptional levels. About 2.0% of chromosomal regions switch from inactive to active (B-to-A) in the mutant line, harboring 58 differentially expressed genes enriched in flavonoid biosynthesis and circadian rhythm functions. The mutant peanut line shows a higher number of genome-wide cis-interactions than its wild-type. The present study reveals a new TAD in the mutant line that generates different chromatin loops and harbors a specific upstream *AP2EREBP*-binding motif which might upregulate the expression of the *GA2ox* gene and decrease active gibberellin (GA) content, presumably making the mutant plant dwarf.

**Conclusions:** Our findings will shed new light on the relationship between 3D chromatin architecture and transcriptional regulation in plants.

**Keywords:** Peanut, 3D structure, Hi-C, ATAC-seq, Gene expression, Gene regulation

## Background

Chromatin interaction and genome organization play an important role in gene expression regulation. Chromatin, the main carrier of eukaryotic genetic information, is folded to confined spatial structure within a preferred but not fixed territory in the nucleus [1, 2]. Lack of information about chromatin spatial organization and chromosome structures in the plant genome have constrained understanding of gene regulation and cellular homeostasis. The recent emergence of high-throughput chromosome conformation capture (Hi-C) technology and Assay for Transposase-

Accessible Chromatin sequencing (ATAC-seq) provided the opportunity to decipher (3D) genome architecture and to dissect the relationship between chromatin organization and gene expression in various biological processes.

Chromatin interaction patterns of mammalian chromosomes have suggested "A" and "B" compartments at the megabase scale, corresponding to euchromatic (gene-rich and highly transcribed) and heterochromatic regions (gene-poor and silent), respectively [3]. Although the A compartment preferentially and spatially clusters in the nuclear interior while the B compartment is near the nuclear periphery [4], neither is static and both can change during cell differentiation in a lineage-specific manner [5, 6]. At the sub-megabase scale, mammalian chromatin can be further divided into topologically associated domains (TADs) [7–10] with their boundaries enriched for architectural proteins such as cohesion and CCCTC-binding factors (CTCF) and specific epigenetic marks [11, 12]. TADs have been shown to be largely conserved across different species, cell types and physiological conditions, and may act as functional units for transcription regulation [7, 13].

Current understanding of the 3D chromatin architecture in plants is mainly derived from Hi-C analyses performed in the model plant *Arabidopsis* [14, 15], and crop plants such as maize (*Zea mays*), tomato (*Solanum lycopersicum*), sorghum (*Sorghum bicolor*), foxtail millet (*Setaria italica*), rice (*Oryza sativa*) [16], and wheat (*Triticum aestivum*) [17]. Although the organization of TADs was not obvious in *Arabidopsis* in contrast to mammals, TADs were observed in plants [16]. Similar study in diploid and tetraploid cotton (*Gossypium* spp.) indicated that 3D genome architecture comprised compartments, TADs and loops, which were further correlated with the expression of homoeologous genes [18]. These previous studies shed light on the relationship between 3D genome organization and transcriptional regulation; however, more such studies are required to elucidate further details and mechanisms of these 3D structures in plant biological functions.

The 3D genome organization regulates gene expression by bringing together distant promoter, enhancer, and other cis-regulatory elements [19]. Chromatin compaction within the nucleus often restricts the access of transcription factors (TFs) to cis-regulatory elements such as promoters and enhancers [20]. Local changes in chromatin properties induced by various mechanisms during cell differentiation could modify the accessibility of regulatory chromatin regions to the transcriptional machinery [21]. This ultimately leads to the establishment of lineage-specific TF regulatory modules and the resulting transcriptional output characteristic of a given cell type. With advances of technology, such as the simple and sensitive ATAC-seq [22], specific TF regulatory modules in plants and research in this area can now be accelerated to detect highly accessible chromatin regions and TF-binding sites within these regions [23].

Plant height is an important trait affecting plant domestication, architecture, lodging resistance, and yield performance [19]. Growth-promoting gibberellins (GAs) are a class of phytohormone that plays a pivotal role in many aspects of plant growth and development, including seed germination, stem elongation, flowering, and plant height [19, 24]. The ability to restore growth of dwarf mutants of pea (*Pisum sativum*) and maize suggested that GAs are endogenous growth regulators in tall plants [25].

Cultivated peanut (*Arachis hypogaea* L.), an important legume crop to provide edible oil, feedstock, and ground cover worldwide, is an allotetraploid (AABB, 2n = 40) with

genomes from progenitors resembling *Arachis duranensis* (AA, 2n=20) and *Arachis ipaensis* (BB, 2n=20). High-quality reference genome sequences for both peanut subgenomes have become available recently [26–28]. Realizing the importance of plant height in peanut yield and lack of understanding of its molecular basis and gene regulation [29], here we compare the genome organization between two cultivated tetraploid peanut lines, wild-type H2014, and its dwarf mutant H1314. We performed RNA-seq of different tissues to obtain whole transcriptome expression profiles, analyzed chromatin accessibility by ATAC-seq, and characterized three-dimensional (3D) genome architecture by Hi-C sequencing. Integrative analyses provided insights into 3D genome architecture and chromatin accessibility in peanut, aiming to reveal multiple layers of coordinated regulation of genes involved in important biological processes in plants.

## Results

### Genome-wide interaction matrix of peanut

To investigate differences in chromatin organization, we performed Hi-C experiments using leaves from wild type H2014 and its dwarf mutant, H1314. A total of 3.0 billion pairs of sequencing reads were generated (Additional file 1: Table S1) and mapped against the reference genome of Tifrunner [26]. Comprehensive sequence analysis identified 272 and 264 million valid interaction read pairs for H2014 and H1314, respectively, for 3D genome construction. We also performed ATAC-seq to study open chromatin regions (Additional file 1: Table S2), and RNA-seq analysis (Additional file 1: Table S3) using rRNA-depleted RNA extracted from tissues including leaf, stem, branch, flower, and seed at three developmental stages (seed 1, seed 2, and seed 3) to facilitate investigation of chromatin topology-mediated transcriptional regulation in dwarf mutant H1314 and wild type H2014.

Intra-chromosomal interactions revealed by Hi-C were much more frequent than inter-chromosomal interactions (Fig. 1A). The frequency of intra-chromosomal interactions decreased with increasing linear distance, and distances from 0 to 400 kb accounted for 80% interactions (Additional file 2: Fig. S1). Surprisingly, the frequency of interactions also increased at very large genomic distances (> 3200 kb). Sophisticated higher-order chromatin structures, including compartments and TADs, were also observed at different length scales (Fig. 1B). Simulated images of the whole genome showed that chromosomes were positioned within confined volumes, which was consistent with the concept of "chromosome territory," i.e., that each chromosome occupies a distinct region exclusive to the nucleus (Fig. 1C). This study also identified several sequence scaffolding errors that could not be experimentally validated (Additional file 1: Table S4), suggesting the opportunity for additional refinement of the cultivated allotetraploid peanut genome sequences assembled from whole-genome shotgun sequencing.

### Widespread A/B compartments in peanut

Hi-C analysis of peanut samples identified A and B compartments with positive and negative eigenvectors, respectively (Fig. 2), similar to mammalian genomes in which these compartments correspond to active and inactive regions, respectively [3]. In both wild type and the dwarf mutant peanut lines, the A compartment with 52.3% of

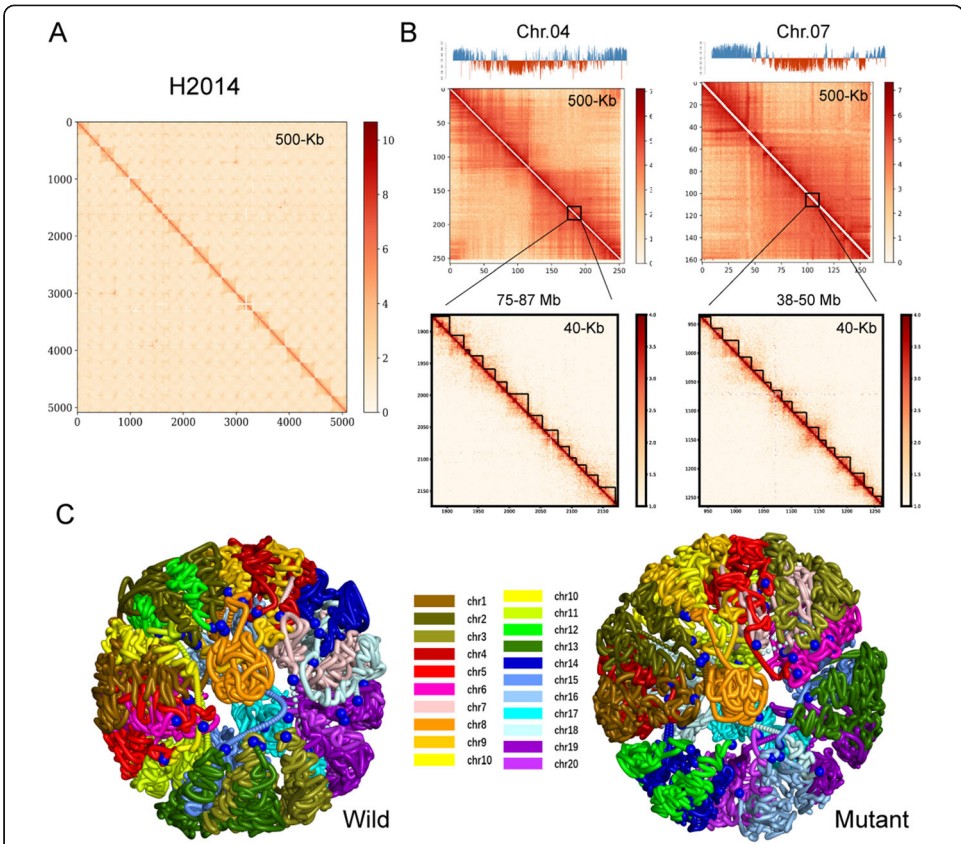

**Fig. 1** Hi-C analyses of chromatin interactions in wild and dwarf mutant peanuts. **A** Genome-wide chromatin interaction map represented by the wild type (H2014) at 500-kb resolution. The chromosomes are stacked from top left to bottom right in order (chr01, chr02…chr20). Color bars beside heat maps indicate strong interactions in red and weak interactions in white. **B** Chromatin interactions represented by a single chromosome of the wild type (H2014) at 500-kb resolution. The upper track shows the partitioning of A-compartments (blue histogram) and B-compartments (red histogram). The middle track shows global patterns of chromatin interaction represented by chr04 and chr07, respectively. The lower track shows chromatin interactions in an enlarged region of chr04 and chr07 at 40-kb resolution, respectively. Each triangle distributed diagonally is represented as a topologically associated domain (TAD). **C** 3D model of whole chromosomes in the wild type (H2014) and the dwarf mutant (H1314). Each color represents one chromosome

genomic regions was bigger than the B compartment (47.7%). As expected, the A compartments showed higher gene density, lower GC content, and significantly higher transcription levels than the B compartments in both lines (Fig. 2A). Most chromosomes had similar distributions of compartments (Fig. 2B), except chr03, chr08, and chr14 that had uneven distribution of A and B compartments (Additional file 1: Table S5; Fig. 2C).

To determine whether there are differences in compartmentalization between the genomes of the wild type and the mutant, we compared the genome-wide organization of A/B compartments at 500-kb resolution. Centromeric regions in the dwarf mutant (H1314) showed different genomic compartmentalization from A to B type and vice versa in comparison with the wild type (Fig. 2C). In total, 2.0% of compartments (50.8 Mb) in the dwarf mutant exhibited B-to-A switching, while 2.1% (51.1 Mb) exhibited A-to-B switching in comparison with the wild type (Additional file 1: Table S6).

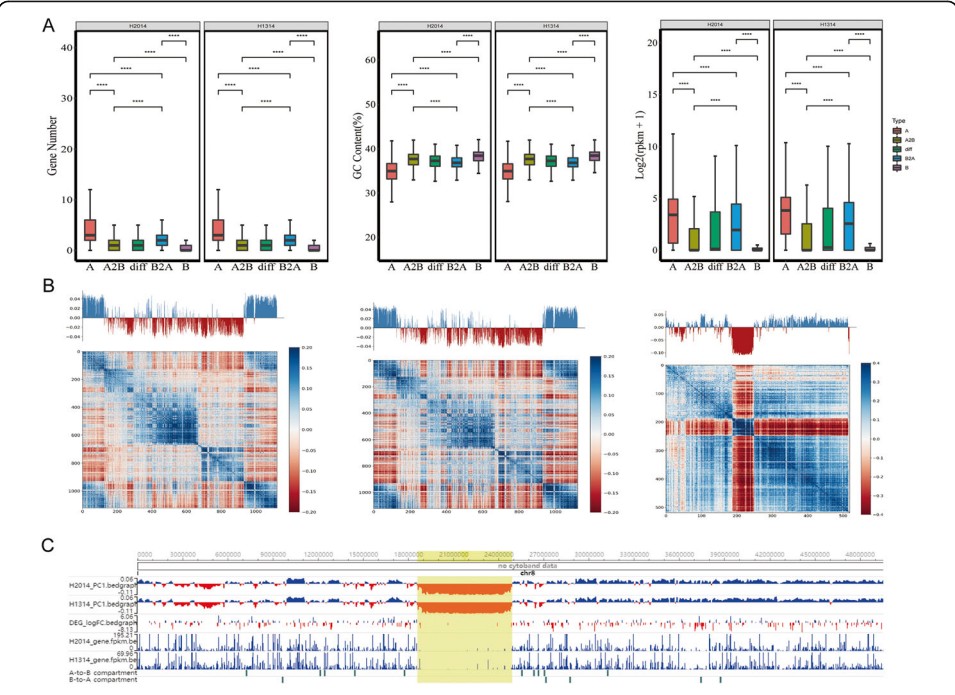

**Fig. 2** Genomic features of A/B compartments in peanut. **A** Illustration of gene density, GC content, and gene expression (FPKM value) of A compartments, A to B switching compartments, different compartments, B to A switching compartments, and B compartments, in the wild type H2014 and the mutant type H1314, respectively. **B** Distribution of A and B compartments represented by chr04, chr07, and chr08 of H2014, with the upper line showing the partition of A (blue histogram) and B (red histogram) compartments. The lower track indicates the first principal component values showing A/B compartment status at 500-kb resolution. **C** The representative genomic region 18.6–24.9Mb on chr08 of H2014 displayed A/B compartments. The top two lanes indicated the first principal component values corresponding to A compartments (blue histogram) and B compartments (red histogram) in H2014 and H1314, respectively. The third lane indicated the DEGs between H2014 and H1314. Blue bars represented the upregulation, and the red bars represented the downregulation. The fourth and fifth lanes indicated the FPKM values of genes in H2014 and H1314, respectively. The remaining two lanes indicated the compartment switching between H2014 and H1314. The yellow shaded region showed uneven distribution of A and B compartments

To explore the impact of compartment changes on peanut gene expression, we first focused on genes residing in genomic regions exhibiting A to B compartment switching in the mutant compared to the wild type. The overall expression patterns of genes were significantly different in regions exhibiting A to B compartment switching than in compartments with conserved status (Fig. 2A). There were 101 and 58 differentially expressed genes (DEGs) identified in genomic regions with A-to-B switching and B-to-A switching, respectively, between the wild type vs the dwarf mutant (Additional file 1: Table S7). KEGG enrichment analyses revealed that DEGs in A-to-B switching were enriched for genes involved in plant-pathogen and mRNA-surveillance pathways (Additional file 2: Fig. S2), while DEGs in B-to-A switching were enriched for genes associated with flavonoid biosynthesis and circadian rhythm-plant.

To clarify the relationship between open spatial compartmentalization and increased gene expression, we analyzed the frequency of DEGs at regions with compartment transition. In total 59 of 101 DEGs (58.4%) showed downregulation when compartment A in the wild type switched to compartment B in the dwarf mutant, while 23 of 58 DEGs (39.7%) showed upregulation when compartment B in the wild type switched to

compartment A in the dwarf mutant. The inconsistent relationship indicated uncoupling of compartment changes and gene expression, similar to results in *Drosophila* [30].

### Chromatin loops in peanut

We identified intra-chromosomal interactions (cis-interaction) in the wild type H2014 (Fig. 3A) and the dwarf mutant H1314 (Additional file 2: Fig. S3). There were 12,661 interactions found in these two peanut lines with contribution of 62.6% (total was 20,368) and 45.9% (total was 27,551) interactions in the wild type and the dwarf mutant, respectively (Additional file 1: Table S8). The total numbers of cis-interactions of A and B subgenomes were 8599 and 11,769 in the wild type, less than the 11,493 and 16,058 in the dwarf mutant, respectively. Among the 10 A-subgenome chromosomes, chr03 had the highest number of cis-interactions in both H2014 (28.9%, 2483/8599) and H1314 (28.4%, 3262/11,493), while chr14 showed a similar proportion of B-subgenome cis-interactions in H2014 (15.2%, 1790/11,769) and H1314 (15.0%, 2407/16,058). Though 60% of significant interactions were short-range (0–400 kb), several chromosomes also displayed obvious long-distance interactions (Fig. 3C).

The allotetraploid nature of peanut permits the formation of inter-subgenomic chromatin interactions (trans-interactions), representing an additional dimension of 3D genome architecture. A total of 7446 trans-interactions were detected in the wild type (Fig. 4B) and the dwarf mutant (Additional file 2: Fig. S3), accounting for 30.5% (of 24,396) and 30.8% (of 24,194) of all trans-interactions in the wild type and the dwarf mutant, respectively (Additional file 1: Table S9). Trans-interactions were grouped into three classes: among A-subgenome (chr01-chr10), among B-subgenome (chr11-chr20), and between A and B subgenomes (Fig. 3F, Additional file 1: Table S10). The proportions of inter-chromosomal interactions among the B-subgenome, 10.6% (2579/24,396)

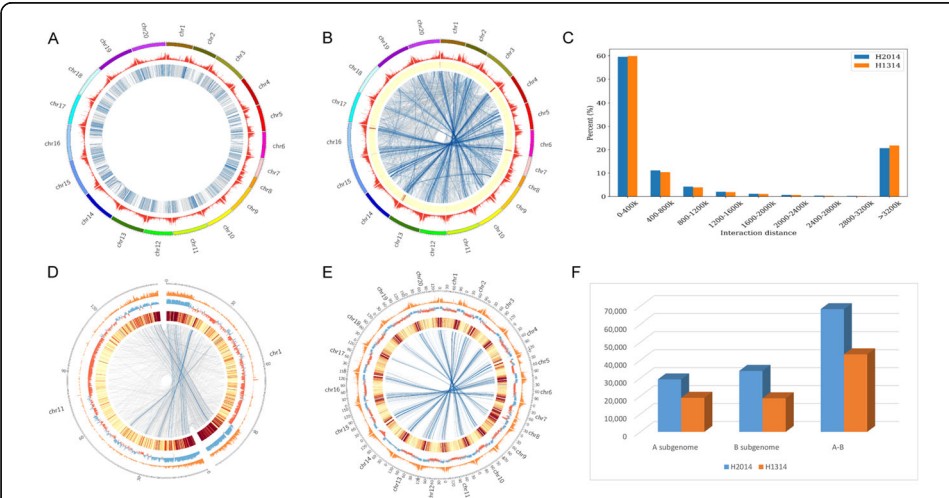

**Fig. 3** Chromatin loops in peanut. **A** Circos of genomic cis-interactions represented by the wild type (H2014). **B** Circos of genomic trans-interactions represented by the wild type (H2014). **C** Distribution of genomic distance of cis-interactions in the wild type and the dwarf mutant. **D** Circos of interactions between subgenome-homoeologous chromosomes represented by chr01 and chr11 in the wild type (H2014). **E** Circos of interactions between homologous blocks of subgenome-homoeologous chromosomes represented by the wild type (H2014). **F** The number of trans-interactions among A subgenome, B subgenome, and between A and B subgenomes in the wild type (H2014) and the dwarf mutant (H1314)

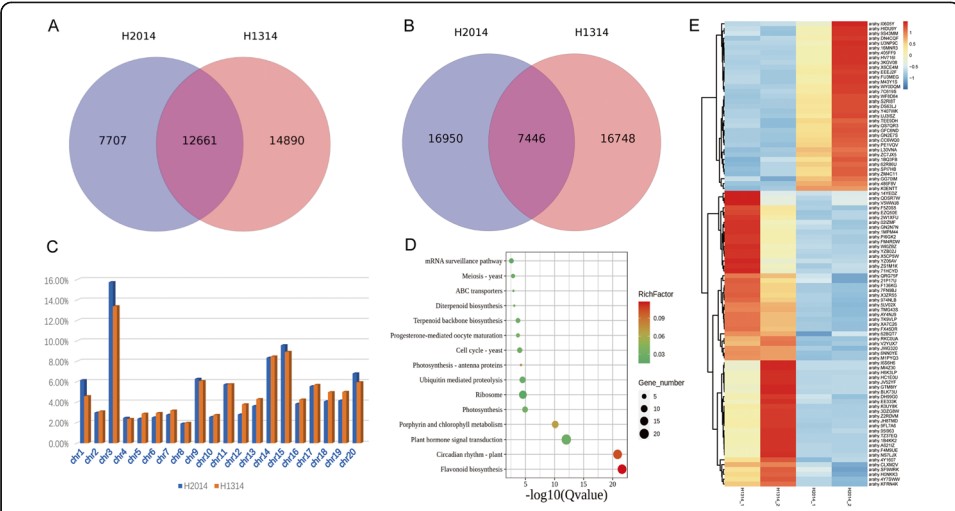

**Fig. 4** Characterization of specific cis-interactions in the wild type and the mutant peanut. **A** Venn diagram for all cis-interactions identified in the two peanut lines. **B** Venn diagram for all trans-interactions identified in these two lines. **C** Distribution of specific cis-interactions of 20 chromosomes. **D** KEGG pathway of 96 differentially expressed genes between the dwarf mutant (H1314) and the wild type (H2014). **E** Heatmap showing the log ratio of normalized FPKM of the 96 differentially expressed genes involved with chromatin loops between the dwarf mutant (H1314) and the wild type (H2014). Each line on the heatmap represents a gene, and the values are given for each of two replicates

in the wild type and 10.0% (2425/24,194) in the dwarf mutant, were higher than those among the A-subgenome (chr1-chr10), 9.6% in the wild type (2350/24,396), and 9.5% in the dwarf mutant (2288/24,194), probably due to the larger size of the B (~1.5 G) than the A-subgenome (~1.2 G). These results suggested that trans-interactions did not tend to be intra-subgenomic.

For an allopolyploid such as peanut, an intriguing question is how the two subgenomes coexist and coordinate interactions and gene regulation. Inter-subgenomic interactions accounted for 79.8% (19,467/24,396) in the wild type peanut, and 80.5% (19,481/24,194) in the dwarf mutant, respectively. Importantly, about 80.8% (15,722/19,467) and 80.0% (15,585/19,481) of inter-subgenomic interactions detected in the wild type and the dwarf mutant, respectively, occur between homoeologous chromosomes (chr01-chr11, chr02-chr12, ......) (Additional file 1: Table S10). Clearly, inter-subgenome-interactions are more frequent between homoeologous chromosomes of allopolyploid peanut.

Trans-interactions between homoeologous chromosomes were significantly enriched at chromosome ends, which were partitioned into the A compartment with high gene density (Fig. 3D). Based on homoeologous genes over the whole genome, we further selected homoeologous blocks and found these blocks were also significantly enriched at the ends of homoeologous chromosomes (Fig. 3E). The prevalence of inter-subgenomic interactions prompted us to explore their possible effects on transcriptional regulation of homoeologous genes (i.e., with a 1:1 counterpart in each subgenome after polyploidization). We compared chromatin interactions in 23,219 homoeologous gene pairs (Additional file 1: Table S11) to their expression, detecting only four homoeologous gene pairs with upregulation and six with downregulation in the wild type (H2014) (Additional file 1: Table S12). Compared with 21.6% and 18.0% homoeologous gene pairs in *Gossypium hirsutum* and *Gossypium barbadense* [18], the low proportion of

homoeologous gene pairs showing significantly different expression levels may reflect a relatively early stage in the divergence of the two sub-genomes in allotetraploid peanut.

### Increased interaction frequency in the mutant is associated with development-related genes

Enhancers frequently control non-adjacent genes over large genomic distances through chromatin looping. We compared all cis- and trans-interactions between the wild and mutant lines. Interestingly, although a larger number of cis-interactions (14,890) were detected in the dwarf mutant than the wild type (7707) (Fig. 4A), there was not much difference in numbers of specific trans-interactions (Fig. 4B). The specific cis-interactions were widely distributed over 20 chromosomes, but enriched in chr03, chr14, and chr15 (Fig. 4C). To investigate the relationship between loops and geno-types, we analyzed the DEGs in these loops. Compared with the wild type, there were 96 genes showing significantly different expression including 61 upregulated and 35 downregulated in the dwarf mutant (Additional file 1: Table S13). KEGG pathway analysis of these genes revealed well known pathways, namely "circadian rhythm-plant" and "flavonoid biosynthesis" (Fig. 4D), important to development in plants. The expression profiles of these genes (Fig. 4E) reveal patterns correlated with aspects of development in peanut.

### Topologically associated domains (TADs) features in peanut genome

The two different compartments (A and B) arise due to associations of topologically associated domains (TADs) that also define the transcriptionally active and inactive chromatin. The 3D maps at 40-kb resolution led to identification of 3353 (boundary number was 3333) and 3363 TADs (boundary number was 3343) by calculating the insulation values in the wild type and dwarf mutant peanut, respectively (Additional file 1: Table S14). Further comparison of chromatin topology between these two lines revealed that 74.4% (2494/3353) of TADs present in the wild type were conserved in the dwarf mutant (Fig. 5A). In addition, other changes for TADs such as 485 merges, 207 splits, and 167 rearrangements were also identified between these two lines. Two regions were selected to represent merges and splits of TADs (Fig. 5B), respectively. These results indicate local chromatin reorganization in the mutant compared to the wild type. We also compared the expression levels of genes residing in conserved and non-conserved TADs—five of the 73 genes in non-conserved TAD genomic regions were differentially expressed between the wild type and mutant lines (Additional file 1: Table S15).

The presence of specific TAD boundaries is crucial for biological functions [11]. Compared with TAD inner regions, TAD boundaries showed higher levels of gene expression (Fig. 5C), gene density (Fig. 5D), and lower GC content (Fig. 5E) in both peanut lines. In contrast to the prevalence of CTCF binding sites at the boundaries in mammals, we found several sequence motifs at the TAD boundaries in peanut (Additional file 1: Table S16). The top two motifs in the wild type were a high-mobility group of proteins (HMG) (76.7%, 2557/3333) and AGL (75.0%, 2499/3333), while in the mutant were HMG (77.2%, 2582/3343) and ARF-2 (74.6%, 2495/3343). These motifs play important roles in plant growth and development [31–33].

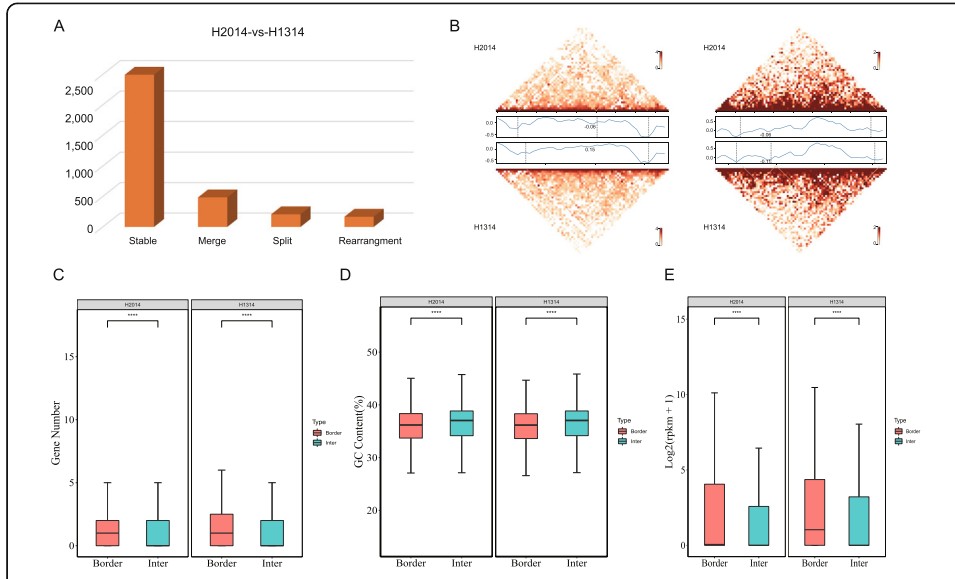

**Fig. 5** Characterization of topologically associated domains (TADs) in peanut. **A** Changes for TAD between the wild type and the dwarf mutant peanuts. **B** Merge of TADs represented by 98480000–100200000 of chr03 and Split of TADs represented by 110840000–112680000 of chr13. The upper and lower heat maps represented H2014 and H1314, respectively. The values on the left indicated the insulation score profile of TADs. The dotted line indicated the TAD border. Comparison of the gene density (**C**), GC contents (**D**), and gene expression (FPKM) (**E**) between TAD borders and inner regions in the wild type (H2014) and the mutant type (H1314)

## ATAC-seq identifies accessible chromatin regions in peanut

We first defined accessible chromatin using ATAC-seq generated from leaves of the wild type and dwarf mutant peanut lines, with two biological replicates. A total of 116.9 and 135.6 million reads were obtained from the raw reads by removing the adaptor sequences (Additional file 1: Table S2), and 99% of all reads were successfully mapped to the peanut reference genome, with 61% and 64% uniquely mapped. For each sample, the fragment size distribution was primarily 100-bp and smaller, indicating that our libraries were composed of primarily nucleosome-free reads (Additional file 2: Fig. S4). Next, we identified local regions of increased accessibility using the MACS2 algorithm for peak calling. As a result, 12,968, 14,974, 20,110, and 20,683 peak sites with cutoff $q$ value < 0.05 were identified in H2014-1, H2014-2, H1314-1, and H1314-2, respectively. The genomic distribution of accessible chromatin peaks was very similar between these two lines, with 86.6% mapped in the intergenic region and 4.4% in promoter-TSS (Additional file 2: Fig. S5). We also calculated the peak distance to the gene transcription start site (TSSs), finding 53% of peaks upstream of TTSs, and 86% of these located outside 2 kb of TSSs (Additional file 1: Table S17).

To examine quantitative differences in accessible chromatin regions between these two peanut lines, we calculated the normalized total read counts at each peak and used DESeq2 to identify quantitative differences in accessibility [34]. Only peaks with $|\log_2 FC| > 1$ and $p$ value < 0.05 were deemed different between these two peanut lines. With this approach, we identified a total of 1805 differentially accessible peaks between the wild type and the dwarf mutant, i.e., 699 peaks with stronger signal and 1106 with weaker signal in the mutant (Fig. 6A).

The presence of specific DNA motifs in promoter regions bound by specific TFs controls the abundance of different mRNAs [35]. To identify DNA motifs that could

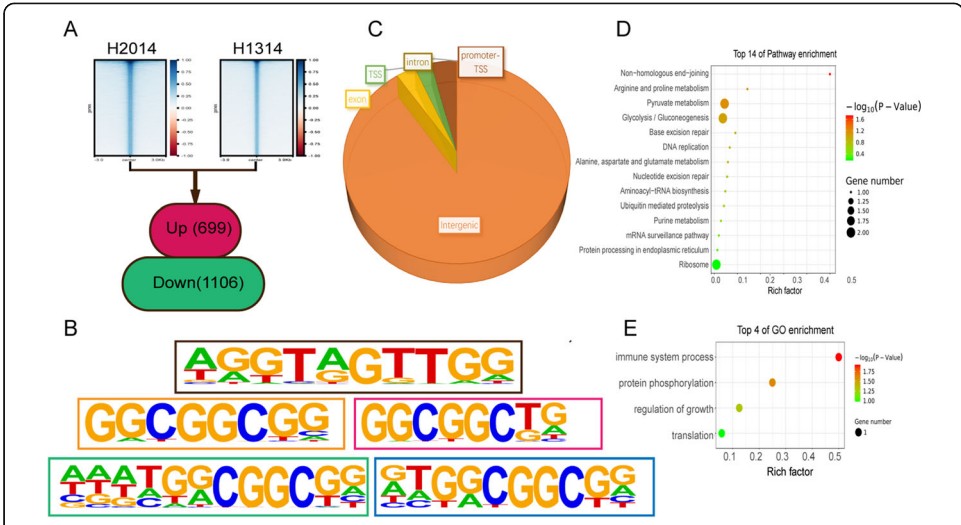

**Fig. 6** Assay for Transposase-Accessible Chromatin using sequencing (ATAC-seq) analysis in peanut. **A** Normalized read signal of ATAC-seq peak sites in H2014, the wild type (left) and H1314, the dwarf mutant (right). Numbers in the bottom box show stronger (red) and weaker (green) ATAC-seq signals in H1314, respectively. **B** The motif sequences identified around the stronger ATAC-seq peaks in H1314, including MYB (top), ESE1 (AP2EREBP) (center left), AT5G23930 (mTERF) (center right), ERF4 (AP2EREBP) (bottom left), and ERF10 (AP2EREBP) (bottom right). **C** Genomic location distribution of increased accessible chromatin signals in H1314. The ratio shows the ascending arrangement as distal intergenic (89.56%), promoter-TSS (4.01%), exon (3.00%), TTS (2.58%), and intron (0.86%). **D** KEGG pathway of the nearest gene located in genomic regions with stronger ATAC-seq signals in H1314. **E** GO enrichment of the nearest gene located in genomic regions with stronger ATAC-seq signals in H1314. Genes in **D** and **E** were identified as the putatively regulated target genes based on that the ATAC-seq peaks were assigned to its nearest transcription start site

perform this function, we performed sequence motif discovery in local regions of 1805 differentially accessible peaks between the wild type and the mutant, finding differences in the motifs most enriched for increased and decreased accessibility in the mutant (Fig. 6B). Interestingly, the most prevalent motif-binding TF, the responsive element-binding protein (AP2/EREBPs) factors, was strongly enriched in increased accessibility of ATAC-seq peaks in the dwarf mutant.

About 90% of the differentially accessible peaks were detected in the intergenic regions (Fig. 6C), the majority located up to 2 kb downstream and upstream of the TSS. This genomic distribution of peaks suggests that the majority of cis-regulatory regions in the peanut genome are located in distal gene core promoters. We also characterized the nearest gene by annotating the binding sites in genomic regions. KEGG pathway analysis of these genes found stronger ATAC-seq peaks in the dwarf mutant, with enrichment in basic biological processes such as pyruvate metabolism and gluconeogenesis (Fig. 6D). Gene Ontology (GO) terms revealed that the genes associated with the dwarf mutant-enriched peaks were involved in "regulation of growth" and "immune system process," which was consistent with the known phenotype of the dwarf mutant (Fig. 6E).

### Linking of DNA regulatory elements to genes predicts interactions relevant to GA biosynthesis

ATAC-seq peaks represent more accessible chromatin regions, which are likely to contain binding sites that can recruit TFs to regulate the expression of nearby genes [36].

We combined RNA-seq libraries generated from leaves with ATAC-seq to assess the relationship between chromatin accessibility and gene expression. We first analyzed the differentially expressed genes between these two lines, including 4062 and 5955 genes showing downregulation and upregulation, respectively, in the dwarf mutant as compared to the wild type. From the DEGs located in genomic regions with changed accessible peaks, we identified 661 and 604 genes in regions of decreased and increased accessibility in the dwarf mutant, respectively. Then, we combined these two results to search for overlapping genes. Compared with the wild type, we identified 35 overlapping genes located in regions with weaker ATAC-seq signal and showing downregulation in RNA-seq, and 65 genes in regions with stronger ATAC-seq signal and showing upregulation in RNA-seq in dwarf mutant (Additional file 2: Fig. S6).

To identify specific TFs that may regulate peanut development, we further identified motif-binding TFs based on sequence motifs that were enriched in both wild type and mutant lines. We then used these TF sets to combine known protein–protein interactions and functional interactions among genes to predict functional connections between a set of upregulated genes and TFs in the dwarf mutant. One ATAC-seq site (chr19: 23,123,339–23,124,121) located in the 9611 bp upstream of TSSs of *Arahy 1SCL5Q* could be detected in the dwarf mutant but not in the wild type. This sequence motif was predicted to bind with one TF, *AP2EREBP*, which is an ancient superfamily of transcription factors, and plays important roles in regulating the development of flowers, ovules, and seeds, and regulating responses to plant hormones (ethylene, ABA, and GA, etc.) [37]. GAs are important regulators of many aspects of plant growth and development, including cell elongation, responses to biotic and abiotic stresses [24]. Upon exploring the putative functional network of AP2EREBP, one of its target genes, *Arahy 1SCL5Q*, located in chr19 (23,133,341–23,140,060) encoded a Gibberellin 2 beta-dioxygenase (GA2ox) which is involved in diterpenoid biosynthesis. GA2ox plays important roles in GA biosynthesis which works as a negative regulator to change active GA into inactive GA [38]. In view of that, GA2ox might be involved in the regulation of plant growth related to the gibberellin signaling pathway. Since the mutant showed significantly dwarf phenotype, we decided to explore this regulatory network in more detail as both TF and GA2ox have been found associated with the regulation of GA, and may, therefore, directly affect the physiology and development of peanut.

### Comprehensive analysis of chromatin architecture and chromatin accessibility involved in the dwarf mutant

To determine whether TAD plays a regulatory role in peanut that resembles its role in mammals [7, 39], we analyzed a genomic region on chr19 (22,950,000–23,360,000), which harbored a binding site for *AR2EREBP* and the *GA2ox* gene (Fig. 7A, B). In this region, four ATAC-seq peaks could be detected in the dwarf mutant and just one in the wild type. The sequence motif which could bind with *AP2EREBP* was located upstream of *Arahy 1SCL5Q*. Compared with the wild type, we found a split of the TAD with different chromatin loops, a new loop (interaction between 23,160,000 and 23,360,000) emerging and one loop (interaction between 23,040,000 and 23,160,000) missing in the dwarf mutant. The chromatin loop involving the GA2ox locus in the wild type was confirmed using quantitative chromosome conformation capture (3C)-

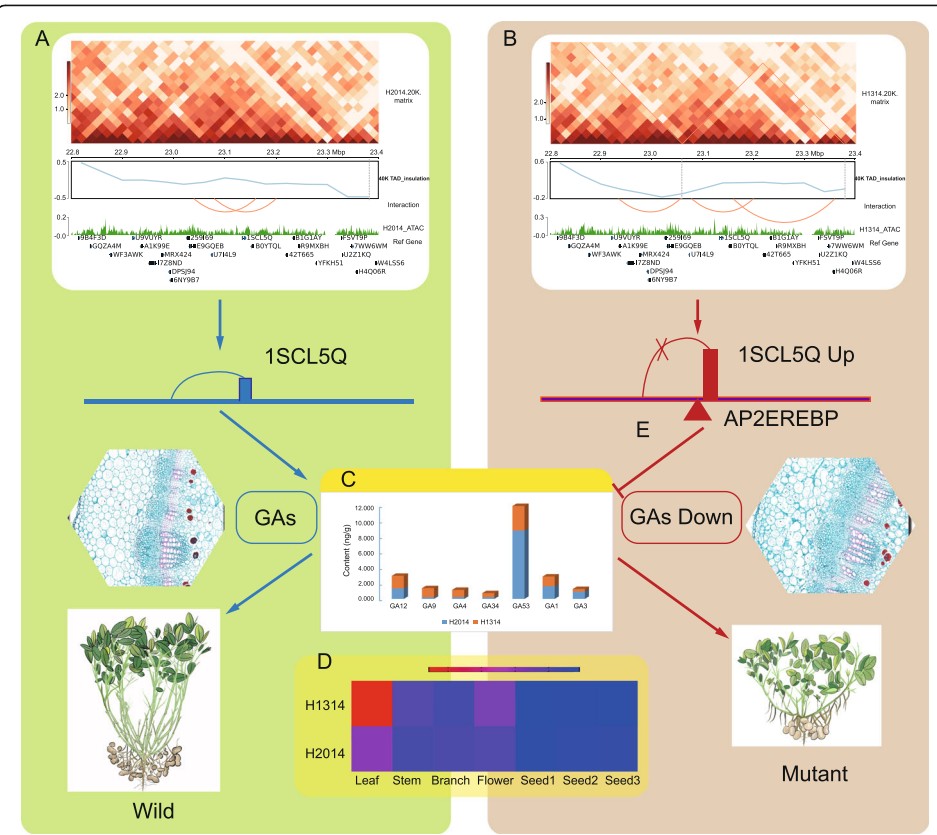

**Fig. 7** The regulatory network of plant height in the wild type (**A**) and the dwarf mutant (**B**). **A**, **B** The snapshot of the Wash U browser view shows chromatin loops and ATAC-seq in the wild type (H2014) and the dwarf mutant (H1314), respectively. Top, Hi-C contact map of a genomic region (22950000–23360000) on chr19; Second lane, insulation scores for TAD; third lane, the TAD regions sorted; fourth lane, loops sorted with red line; fifth lane, ATAC-seq peaks; bottom, gene annotation in the genomic region. **C** The column shows the relative content of GA in different tissues. The histological sections shown on both sides revealed the regular and irregular arrangement of cells in wild type and the mutant, respectively. **D** Heat map of *GA2ox* gene in different tissues. The regulatory pattern of endogenous GA levels by the Gibberellin 2 beta-dioxygenase (encoded by 1SCL5Q) through interacting with AP2EREBP. The phenotype of two peanut types shows the obvious differences in plant height

PCR experiments (Additional file 2: Fig. S7). qPCR results showed the relative chromatin loop frequency was significantly enriched with the anchor primer set in the cross-linked samples, suggesting that the chromatin architecture are important to regulate *GA2ox* expression and eventually influence the endogenous GA and the plant height.

The dwarf mutant, H1314, had shortened internodes and obviously reduced height (Fig. 7). The tissue section of the main stem showed regular arrangement of cells in the wild type, H2014, but irregular arrangement of compartment cells in the dwarf mutant. To assess the effect of GA content on plant height, levels of seven endogenous GAs (GA12, GA9, GA4, GA34, GA53, GA1, and GA3) were determined in the leaves of the wild and the dwarf mutant (Fig. 7C). Bioactive GA4 was highly increased in the dwarf mutant as compared to non-detectable change in GA1. GA12 and GA53, the substrates for GA2ox, decreased to similar levels relative to the wild type. Unexpectedly, GA9 and GA34 were increased in the dwarf mutant, while levels of bioactive GA3 were increased in the wild type. We also compared the expression levels of *GA2ox* of these two peanut lines (Fig. 7D), and the results showed that *GA2ox* in the dwarf mutant was

significantly higher than that of the wild type in leaf, stem, and flower tissues. There was no obvious difference of expression levels between these two lines in other tissues, including branches and seeds at three different development stages.

## Discussion

The spatial organization of chromatin plays critical roles in regulating gene expression. Lack of understanding of 3D genome architecture limits progress in plant gene manipulation strategies. A series of Hi-C analyses in the model plant *Arabidopsis* partitioned chromosomes into A/B compartments but could not detect TAD domains, while such information was completely retrieved in the mammalian genome [14, 15, 40]. A similar study in cotton reported the first concrete evidence of plant genome partitioning of A/B compartments as well as TADs [18]. Hi-C studies in hexaploid wheat also revealed TAD-like domains, the boundaries of which coincide with high transcriptional activities and active epigenetic architecture [17]. Considering the complexity posed by the allopolyploid genome of cultivated peanut, understanding of genome architecture and its impact on gene expression regulation will facilitate better utilization of available genome biology for genetic improvement [41]. In this study, we revealed the 3D chromatin organization of peanut for the first time, finding compartmentalization consistent with the respective compact and loose structural domains in chromosomes. However, we did not observe intense cis-interaction signal on the anti-diagonal lines between the two chromosome arms or the intense trans-interaction between the centromere regions in different chromosomes, which were reported in two crops with large genomes, maize (2.4G) [16], and barley (5G) [42]. The genome-wide intra-chromosomal interactions displayed the expected reduction of contact probability as a function of increasing genomic distance in both wild type and mutant lines, as found in most previous Hi-C studies. Surprisingly, very distant regions showed a higher intra-chromosome interaction frequency in peanut, which might due to the folding status of the corresponding chromatin.

Increasing evidence highlights the importance of chromosome architecture in regulation of gene expression of important biological processes in mammals [7, 13, 39]. In peanut, we observed widespread A/B compartment switching, TAD changes, and interaction frequency in both peanut lines, similar to tetraploid cotton [18], but not diploid *Arabidopsis* with relatively small genome size. These results indicated that complicated genomes tended to diverge into A and B (compact or relaxed) compartments and evolve TADs as a dosage compensation mechanism to balance homologous gene expression. As a result, low gene activities in compacted genomic regions could facilitate neofunctionalization, serving as a genome evolution reservoir. The DEGs identified between the wild type and mutant were involved in different biological processes, suggesting that chromatin structural changes do regulate gene expression. Without CCCTC-binding factors (CTCF), several motifs around the TAD boundaries were found in the peanut genome, but there was no further functional characterization of these elements mediating the 3D genome landscape in peanut. Further studies should explore the putative roles of motifs in TAD organization in the peanut genomes.

Chromatin compaction within the nucleus often restricts the access of transcription factors (TFs) to cis-regulatory elements such as promoters and enhancers [20]. Mapping transposase-hypersensitive sites allows for detecting highly accessible chromatin

regions and subsequent identification of TF-binding sites within these regions [23]. In this study, most of the ATAC-seq peaks were located in intergenic regions, colocalized extensively with regulatory elements such as enhancers and promoters. These regions are known to display dynamic chromatin accessibility to induce stage-specific expression of downstream genes [35]. In addition, the significant number of peaks located at least 2 kb away from TSS of any reference gene, indicated that distal regulatory regions were also detected. The cluster-specific peak sets were enriched for motifs of TFs with correlated gene expression which have been known to play important roles in plant development. Different sequence motifs were enriched within the ATAC-seq peaks, and their binding TFs were also classified in the wild type and dwarf mutant of peanut.

During differentiation, cells employ various mechanisms to induce local changes in chromatin properties, thereby modifying the accessibility of regulatory chromatin regions to the transcriptional machinery [20, 21]. This allowed us to identify TFs that are likely to bind at these regulatory elements and to construct specific TF regulatory networks. Over-expression of the salinity-responsive *DWARF AND DELAYED FLOWERING 1* (*DDF1*) gene, encoding an AP2 transcription factor, causes dwarfism mainly by reducing levels of bioactive GA in transgenic *Arabidopsis* [38]. Transient overexpression of *DDF1* activated the *GA 2-oxidase 7* (*GA2ox7*) gene, which encodes a C20-GA deactivation enzyme in *Arabidopsis* leaves. These results demonstrate that *Arabidopsis* plants actively reduce endogenous GA levels via the induction of GA 2-oxidase leading to growth repression for stress adaptation. From these findings, combining the irregular arrangement of cells and specifically enriched motifs for a TF, *AP2/EREBP*, in the dwarf mutant, we strongly suggest that the regulatory network involved with *GA2ox* is an important mechanism controlling peanut plant architecture. In this study, we searched for *AP2/EREBP* target gene(s) responsible for decreasing bioactive GA levels, finding *Arahy 1SCL5Q* on chr19 encoding GA2ox. The upregulation of *GA2ox* could bind with *AP2/EREBP* with the help of its upstream sequence motif leading to reduced GA content in the dwarf peanut mutant. Over-expression of *GA2ox* in *Arabidopsis* reduced GA content and resulted in a dwarf phenotype, reduction of pollen tube, extension of flowering time, and seed sterility [43–45]. The similarity in gene regulation of stature phenotypes in plants as divergent as *Arabidopsis* and peanut suggests that these findings may also be applicable to other crop plants.

High throughput next-generation sequencing (NGS) technology has allowed to obtain massive amounts of genomic sequences, but it is not sufficient to produce reference-quality genomes [46]. Extensive rearrangements in *Drosophila* genome were reported to cause many changes to chromatin topology, disrupting long-range loops, TADs [30]. Recently, Hi-C was used to solve outstanding challenges related to genome assembly [47–49]. In this study, several scaffolding errors were also identified, such as inversions on chr5 and chr11, and inter-chromosomal translocation between several homoeologous sub-genomes. These errors would influence the number of TADs and A/B compartments, but could not influence the main results. Integration of Hi-C, RNA-seq, and ATAC-seq data produced in the same lines is helpful to reveal new insights into the relationship between chromosome conformation and gene regulation [45]. Here, we speculated that structure changes and different chromatin accessibility affecting regulatory networks influenced the height of the dwarf mutant. The discovery of a new TAD in the dwarf mutant generated different chromatin loops, which overlapped with an

important region containing a key regulator of GA biosynthesis, *GA2ox*. Chromatin loops have been reported to be mediated by TFs and function in specific gene expression regulation in plants [50–52]. A chromatin loop at the *WUSCHEL* (*WUS*) was found to repress *WUS* expression during flower development in Arabidopsis [53]. The changed architecture increased chromatin accessibility in this region, which harbored a specific TF-binding motif. The TF, *AP2EREBP*, specifically binds with the promoter or enhancer upstream of *GA2ox*, upregulating expression. The high expression of *GA2ox* in turn negatively regulated GA biosynthesis, reducing the active GA contents and resulting in the dwarf phenotype of the mutant. Though more experimental studies are needed to explore the biological functions of topology changes and chromatin accessibility by linking them to important agronomic traits in plants, the results will expand our current understanding of regulatory functions during plant development.

## Conclusions

The integration of the Hi-C method with ATAC-seq enabled reconstruction of 3D genome architecture maps and location of chromatin regions differentially accessible between wild-type and dwarf peanut lines and containing important cis-regulatory motifs, which may help further in developing similar understanding of other traits in other plants. This initial effort opens up a new area of research for peanut and other plant researchers to explore and understand genome compartmentation in gene regulation affecting a wide range of traits. In addition, new data types and their analysis provided new insights and will serve as a valuable resource to derive further models of transcriptional regulatory networks relevant in plants.

## Methods

### Plant material and sample collection

Two peanut lines, wild type H2014 and its dwarf mutant H1314, were planted in the Experimental Station of Henan Agricultural University, Zhengzhou, China. The wild-type peanut line H2014 is a Spanish type peanut with normal plant height. After EMS treatment, a dwarf mutant, H1314, was selected at the 10th generation. Both wild and mutant lines were grown in the greenhouse with three replications. Fresh leaves were collected at the two fully expanded leaf stage, also collecting stem, branch, flower, and seed tissues at corresponding stages. All these tissues were immediately washed twice using sterile water for RNA sequencing (RNA-seq). The leaf tissues were collected for Hi-C analysis and ATAC-seq. Stems were collected for cytological observation of characteristics of cells, including the number and length of cells, diameter, and arrangements of cells using electron microscope between the wild type and the dwarf mutant.

### Measurement of gibberellin content

Leaves (100 mg fresh weight) of the wild type H2014 and dwarf mutant H1314 were collected for phytohormone analysis according to the procedures reported in Xin et al. [54]. After removal, the samples were placed in liquid nitrogen immediately and then ground to a fine powder using a MM-400 milling mixer (Retsch, Haan, Germany). Endogenous gibberellins were extracted and purified with a tailored solid phase extraction procedure based on their physicochemical properties and then analyzed by UPLC-MS/

MS (Waters, Milford, MA, USA). Instrument control and data acquisition and processing were performed using Analyst 1.6.2 software (AB SCIEX, Foster City, CA).

### Hi-C library construction and sequencing

The leaf tissues of three replications were sampled and mixed for Hi-C analysis. About 2.0 g clean leaves were cut into 1–2 mm strips for Hi-C library construction. The nuclear DNA was digested using 200U MboI (NEB) at 37°C for 2 h. Restriction fragment ends were labeled with biotinylated cytosine nucleotides using biotin-14-dCTP (Tri-LINK). Blunt-end ligation was carried out at 16°C overnight in the presence of 50 Weiss units of T4 DNA ligase. After ligation, cross-linking was reversed by 200 μg/mL proteinase K (Thermo) at 65°C overnight. According to the manufacturers' instructions, DNA purification was achieved through QIAamp DNA Mini Kit (Qiagen), and then, purified DNA was sheared to a length of ~400 bp. Point ligation junctions were pulled down by Dynabeads® MyOne™ Streptavidin C1 (Thermofisher), and the Hi-C library for Illumina sequencing was prepped by NEBNext® Ultra™ II DNA library Prep Kit for Illumina (NEB) according to the manufacturers' instructions. Fragments between 400 and 600 bp were paired-end sequenced on the Illumina HiSeq X Ten platform (San Diego, CA, United States) with 150PEmode.

### Construction of fine scale contact map

After quality filtering using Trimmomatic (version 0.38), the clean Hi-C data was mapped to the reference genome [26] (http://www.peanutbase.org) using the Juicer software [55]. Dangling-ends and other unusable data were filtered. The valid pairs of sequences were pooled together for further analysis into 500-kb, 100-kb, and 40-kb non-overlapping genomic intervals, respectively, to generate contact maps [10]. The map resolution is meant to reflect the finest scale at which one can reliably discern local features. The contact maps were normalized by using HiC-Pro software (version 2.7.1).

### Identification of A and B compartments

Compartments are defined as groups of domains, located along the same chromosome or on different chromosomes that display increased interactions with each other. Principal component analysis (PCA) readily differentiates A or B compartments that tend to be captured by the first component. For each arm on an individual chromosome, genomic bins with a positive or negative first eigenvector (PC1) were divided into the A or B compartments. The active "A" compartments are gene-dense euchromatic regions, whereas the inactive "B" compartments are gene-poor heterochromatic regions.

### Analysis of topologically associated domains (TADs) and motif

TADs are contiguous regions that display high levels of self-association and which are separated from adjacent regions by distinct boundaries. The locations of TADs can be determined when interactions occur within 40 kb bins. Locations and numbers of TADs for each sample were identified by using an insulation score algorithm [56]. Motif calling was analyzed on the whole genome using the MEME software, and all motifs were filtered with $q$ value < 0.0001 and $q$ value < 0.001. The TAD boundaries

were identified by calculating the insulation plot of the 40 kb resolution genome-wide interaction maps and named each bin on both side of one TAD as the border for calculating the enrichment of motifs.

### Calculation of intra-and inter-chromosome interactions

The contacts between 10 Kb bins of intra-chromosome and inter-chromosome interactions of each sample were transferred to Ay's Fit-Hi-C software (v1.0.1) to calculate the corresponding cumulative probability $P$ value and false discovery rate (FDR) $q$ value [57]. After calculation, the interactions in which both the $P$ value and $q$ value were less than 0.01, and contact count > 2 were deemed significant.

### ATAC-Seq library preparation and data processing

We prepared ATAC-seq libraries from leaves for each peanut line with two replications to identify open chromatin regions relevant to our experimental traits. Chromatin from intact nuclei was fragmented and tagged following the standard ATAC-seq protocol [22]. Libraries were purified using Qiagen MinElute columns before sequencing. Libraries were sequenced as paired-end 51-bp reads on an Illumina HiSeq2500 instrument.

We used Bowtie version 2.2.3 to align the reads to the reference genome of peanut Tifrunner [26]. For downstream analysis, we removed PCR duplicates using samtools rmdup and required alignment quality scores >30. This step resulted in a significant reduction in the number of reads, as many originated from redundant regions of the chloroplast genome or from nucleus-encoded chloroplast genes. The final number of aligned reads was used for downstream analysis.

To compare the ATAC-seq samples to each other with respect to location and number of ATAC-seq cut sites (first base of an aligned fragment and first base after the fragment), we counted the number of cuts in all non-overlapping windows of 1000 bp in each library. For each pair of libraries, we then calculated Pearson correlations of numbers of cuts (in log space after adding a pseudo count). In order to define an atlas of accessible regions to be used in network inference, we combined the ATAC-seq results from all libraries to maximize the number of identified nucleosome-free regions in the genome relevant to our experimental traits. To define open regions, we counted the number of ATAC cut sites that fell into the 72-bp window centered on each base. We considered a base open if its window contained at least one cut site in more than half of the libraries. If two open bases were less than 72 bp apart, we called all intermediate bases open.

We analyzed differential accessible peaks between the mutant and wild type through 3 steps, i.e., (1) merging the peak files of each sample using the bedtools software, (2) counting the reads over the bed for each sample using bedtools multicov, and (3) assessing differentially accessible peaks using DESeq2. The region was called differentially accessible if the absolute value of the $\log_2$ fold change > 1 at a $p$ value < 0.05.

### Sampling and sequencing for RNA-seq samples

The total RNA of all tissues used in this study was extracted using a guanidine thiocyanate method. Libraries were constructed for two replications using an Illumina TruSeq RNA Library Preparation Kit and sequenced on an Illumina HiSeq 3000 system.

The clean sequencing data were mapped against the reference genome using Tophat2 with default settings [58]. The Cufflinks program (version 2.2.1) was employed to calculate the expression level for each gene. The genes differentially expressed between the mutant and wild type lines were identified using the DESeq package with the negative binomial distribution (FDR < 0.05).

### GO enrichment and KEGG pathway analysis

The reference proteome of peanut was obtained from the Uniprot Database. The Gene Ontology resource was then queried by these IDs, returning all annotations attributed to genes in the reference proteome. Any protein with an annotation to a GO term also gains annotations for all terms that are parents of the given term, as specified by the GO hierarchy. BLASTp was performed, using default parameters, and for each locus ID from the RGAP, the best-matching Uniprot ID was chosen, and the annotations transferred from that Uniprot ID to the locus ID. Enrichment analysis of predictor targets was performed using the GO stats R package, where all genes present in the network were used as background universe.

### 3C experiments

3C experiments were constructed according to previous studies [53, 59, 60]. Briefly, samples were cross-linked under vacuum infiltration for 30 min with 3% formaldehyde at 4°C and quenched with 0.2 M final concentration glycine for 5 min. The cross-linked samples were subsequently lysed. Endogenous nuclease was inactivated with 0.3% SDS, then chromatin DNA were digested by 100 U H*ind*III (NEB) and ligated by 50 U T4 DNA ligase (NEB). After reversing cross-links, the ligated DNA was extracted through QIAamp DNA Mini Kit (Qiagen) according to manufacturers' instructions. The uncross-linked samples were used the same experimental procedure to obtain ligated DNA. Those ligation products were quantified by qPCR in combination with primers (Additional file 1: Table S18) specific for potential interaction sites to detect relative interaction frequency with three biological repeats. *ACTIN7* served as the internal control.

## Supplementary Information

---

**Additional file 1.** Supplementary tables (Table S1 to Table S18).

**Additional file 2.** Supplementary figures (Fig. S1 to Fig. S7).

**Additional file 3.** Review history.

---

### Acknowledgements

The authors are grateful to the anonymous reviewers for their helpful suggestions about the manuscript. The work reported in this article was undertaken as a part of the CGIAR Research Program on Grain Legumes and Dryland Cereals (GLDC). ICRISAT is a member of the CGIAR.

### Review history

The review history is available as Additional file 3.

### Peer review information

## Authors' contributions

D.Y. and R.K.V. conceived and designed the study. K.Z., X.M., Z.L., K.Z., and F.G. performed the experiments. D.Y. provided the mutants. X.G. and M.K.P. summarized and interpreted results, and wrote the paper. D.Y., J.W., B.G., and R.K.V. reviewed and edited the manuscript. The authors read and approved the manuscript.

## Funding

This work was financially supported by grants from the National Natural Science Foundation of China (No. 31471525) for the design and execution of the study. Grants from Key Program of NSFC–Henan United Fund (No. U1704232) and Key Scientific and Technological Project in Henan Province (No.201300111000; S2012–05-G03) for variety breeding and seed preservation. It was also funded in part through Innovation Scientists and Technicians Troop Construction Projects of Henan Province (No.2018JR0001) for data acquisition, data analysis, and writing.

## Availability of data and materials

The primary data generated in this study is available in the Sequence Read Archive (SRA) database (Accession ID: PRJNA430760; https://www.ncbi.nlm.nih.gov/sra/?term=PRJNA430760) [61]. All the secondary datasets pertaining to the present study has been submitted as Additional files with this manuscript.

## Declarations

### Ethics approval and consent to participate

Not applicable

### Consent for publication

Not applicable

### Competing interests

The authors declare that they have no competing interests.

### Author details

[1]College of Agronomy, Henan Agricultural University, Zhengzhou, China. [2]International Crops Research Institute for the Semi-Arid Tropics (ICRISAT), Hyderabad, India. [3]University of Florida, Gainesville, USA. [4]Crop Protection and Management Research Unit, USDA-ARS, Tifton, USA. [5]State Agricultural Biotechnology Centre, Centre for Crop and Food Innovation, Food Futures Institute, Murdoch University, Murdoch, Western Australia, Australia.

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

## 

