## [**Additional file 3.** Review history. · Genome Biology]

Review History

First round of review

Reviewer 1

Are you able to assess all statistics in the manuscript, including the appropriateness of statistical tests used? Yes, and I have assessed the statistics in my report.

Comments to author:

Zhang et al. presented the first Hi-C data in wild type and mutant allotetraploid peanut lines. The most interesting finding is the split of TAD in mutant line which possibly contributed to the phenotypic difference. However, more analysis and experimental validation are needed to support the conclusion.

Major:

1. Page 17 Line 443-444 mentioned "Both wild and mutant lines were grown in the greenhouse with three replications.", but there is only one set of Hi-C data in Table S1.
2. The result about the disruption of TAD in mutant is interesting, which needs further support.
 - 1) Further data analysis. Whether the finding of TAD split is caused by differential mapping due to sequence divergence between WT and mutant instead of true merge and split.
 - 2) Experimental validation, including another replicate or 3C, in order to ensure the repeatability of the main results (Fig7A, B).
3. The conclusion in the Abstract need to be revised. The association between TAD split and GA2ox upregulation is not necessarily decisive, which still need genetic experimental validation.
4. Page 7 Line171-183 and Fig3F: the statistics of interaction need to be normalized in order to be comparable, taking into account the number of interacting chromosomes and chromosome length. In addition, most conclusions regarding enrichment (e.g. P8L212) need to be deduced based on normalized statistics or interacting density instead of the number of interactions.
5. Fig5B is not a good example of merge and split. The difference is not as obvious as Fig7. It may be caused by differential mapping due to sequence divergence between WT and mutant instead of true merge and split. In addition, the figure does not indicate what does the upper/lower heat maps and the left/right heatmaps refer to, nor does it explain what the blue and dotted lines between the heat maps refer to.
6. The resolution of all figures are too low.
7. All legends need more detailed explanation.
8. More detailed statistics of RNA-seq data need to be demonstrated. The data quality, the mapping result, the comparison result, and the repeatability between replicates.

Minor.

1. P3L62-64, recently published Hi-C data in wheat need to be cited.
2. Page6 Line136-138 and Fig2C are to illustrate "Centromeric regions in the dwarf mutant (H1314) showed different genomic compartmentalization from A to B type and vice versa in comparison with the wild type". Are all centromeric regions shown in the figure, or is it the yellow shaded region?
3. The legends of Fig. 4A and 4B are the same. It seems one is cis- and the other is trans-.
4. Page8 Line192-204, homoeologous region has more interactions, but the expression difference of homoeologous genes is very small. Is the intensity of homoeologous region interaction and expression difference correlated?

Reviewer 2

Are you able to assess all statistics in the manuscript, including the appropriateness of statistical tests used? Yes, and I have assessed the statistics in my report.

Comments to author:

In the manuscript, Zhang and colleagues applied multi-omics methods, such as Hi-C, RNA-Seq and ATAC-seq, to study the difference between wild type and mutant peanut lines, especially about the 3D genome structure. They found that there are changes in the A/B compartments, TADs and interactions between wild type and mutant peanut lines, and some motifs and differential expressed genes are associated with open chromatin regions and chromatin structure changes. Although such findings are observed at the first time in peanuts, conceptually they have been reported in other species, such as cotton. Before the manuscript is considered for publication in *Genome Biology*, some issues need to be addressed.

Some major concerns are as follows:

1. For 3D genome analysis, one main issue is the quality of the reference genome assembly. As the authors mentioned in Line 119-120: "This study also identified several sequence scaffolding errors that could not be experimentally validated". Such kind of scaffolding errors will affect the identification of 3D genome structure from Hi-C in the downstream analysis. The authors need to discuss this in the manuscript.
2. For allotetraploid genomes, one issue is the homoeologous sequences in the subgenome-homoeologous chromosomes. In Lines 184-204 and in Figure 3D-F, the authors discussed the trans-interactions between homoeologous chromosomes. The homoeologous sequences in the subgenome-homoeologous chromosomes will affect the mapping results and the identification of chromatin interactions. The authors need to discuss the possibility of mis-mapped reads from one homoeologous chromosome to another homoeologous chromosome. Such mis-mapping of the reads from the homoeologous chromosomes could affect the identification of the trans-

interactions between homoeologous chromosomes.

3. In Figure 1, TADs are not clear, although there are marked triangles in the heatmaps.
4. In Figures 1 and 2, it is better to compare the heatmaps from the wild-type and mutant. It is interesting to know whether there are differences in the 3D genome structure between the wild-type and the mutant.
5. Figure 2: How to define "different compartments"?
6. In Fig 5B, I cannot see some difference in the heatmap, although some lines are marked near the heatmap.
7. The "conclusions" in the abstract is more like a part of the results, not a real conclusion.
8. The bin size in TAD analysis is 50 kb, which is quite big for motif analysis (lines 241-244). Need some detailed explanation for the motif analysis here.

Some minor concerns are as follows:

1. For Figure 3B, the legend is "(B) Cis-interactions identified in wild type and dwarf mutant lines". Figure 3B is more like trans-interactions in one peanut line, either wild type or mutant line.
2. In Fig 4, the figure legend is as follows:
"(A) Venn diagram for all cis-interactions identified in the two peanut lines.
(B) Venn diagram for all cis-interactions identified in these two lines."
Is there something wrong in these two legends? (B) should be about trans-interactions, right?
3. Figure 7C is blur.
4. There are inconsistencies between the TAD resolution in the method part (lines 494-498) and the TAD resolution in the main text (line 242). Need to clarify it.

Authors Response

Point-by-point responses to the reviewers' comments:

General comment from the Reviewer#1: Zhang et al. presented the first Hi-C data in wild type and mutant allotetraploid peanut lines. The most interesting finding is the split of TAD in mutant line which possibly contributed to the phenotypic difference. However, more analysis and experimental validation are needed to support the conclusion.

Authors' Response: Authors are thankful to the Reviewer#1 for appreciating the findings of this MS and the suggestions which helped us in improving this MS further. As suggested, we have also performed validation through 3C assay about the key chromatin loop on chr19 (interaction between 23,040,000 and 23,160,000) for which qPCR results also confirmed solid evidence for the chromatin interaction. All these results have been added in the revised manuscript and provided as Supplementary Fig.7 and Table S18.

Comment 1 of the Reviewer#1: Page 17 Line 443-444 mentioned "Both wild and mutant lines were grown in the greenhouse with three replications.", but there is only one set of Hi-C data in Table S1.

Authors' Response: We thank the Reviewer for seeking clarity on this aspect. This study used two peanut lines (wild type H2014 and its mutant type H1314) for comprehensive analyses. Both wild and mutant lines were planted and sampled in three replicates, and the mixed samples for each line were used for Hi-C sequencing based on their genetic background. Most previous

studies in mammalian genomes (Rasim Barutcu et al., 2015) and plants (Wang et al., 2018) combined the Hi-C data from two replicates for further analyses. We added the corresponding information about replicates for Hi-C, ATAC-seq and RNA-seq, respectively, to clarify the samples in the revised manuscript.

Comment 2 of the Reviewer#1: The result about the disruption of TAD in mutant is interesting, which needs further support. 1) Further data analysis. Whether the finding of TAD split is caused by differential mapping due to sequence divergence between WT and mutant instead of true merge and split. 2) Experimental validation, including another replicate or 3C, in order to ensure the repeatability of the main results (Fig7A, B).

Authors' Response: We thank the Reviewer for these important suggestions.

(1) In our study, in order to characterize different scales of genome organization, we performed comprehensive analyses between the wild type H2014 and its mutant type H1314. Hi-C data was analyzed using Trimmomatic (version 0.38) for quality filtering, then the clean Hi-C data were mapped to the reference genome sequence using the bowtie 2 software (version 2.3.3). For each line, contact maps were constructed with the pipeline HiC-Pro software (version 2.7.1) with default settings. All the interaction frequency were performed using the packages HiTC v1.26.0 and ggplot2 v3.1.0. Single-chromosome Hi-C interaction heatmaps were produced with HiCPlotter v0.8.1, and the TopDom package was applied to identify TADs in the peanut genome. Based on the previous studies in mammalian genomes (Dixon et al., 2012; Rao et al., 2014) and plants such as cotton (Wang et al., 2018) and wheat (Jia et al., 2021), the TAD split between WT and mutant was presumed to be a true split.

(2) Based on the suggestion of Reviewer#1, we have performed 3C assay to validate the key chromatin loop on chr19 (interaction between 23,040,000 and 23,160,000). The qPCR results provided solid evidence for the chromatin interaction. All these results have been added in the revised manuscript and provided as Supplementary Fig.7 and Table S18.

Comment 3 of the Reviewer#1: The conclusion in the Abstract need to be revised. The association between TAD split and GA2ox up-regulation is not necessarily decisive, which still need genetic experimental validation.

Authors' Response: Many thanks for your kind suggestions. We have now revised the corresponding description about the conclusion in the Abstract.

Comment 4 of the Reviewer#1: Page 7 Line171-183 and Fig3F: the statistics of interaction need to be normalized in order to be comparable, taking into account the number of interacting chromosomes and chromosome length. In addition, most conclusions regarding enrichment (e.g. P8L212) need to be deduced based on normalized statistics or interacting density instead of the number of interactions.

Authors' Response: We greatly appreciate for sharing this observation. In our study, Hi-C data was analyzed using Trimmomatic (version 0.38) for quality filtering, then the clean Hi-C data were mapped to the reference genome sequence using the bowtie 2 software (version 2.3.3). For each line, contact maps were constructed with the pipeline HiC-Pro software (version 2.7.1) with default settings. All the interaction frequency was performed in the peanut genome using the packages HiTC v1.26.0 and ggplot2 v3.1.0, which is similar to the previous report in cotton (Wang et al., 2018). All the inter-chromosomal interactions (trans-interactions) between two

chromosomes were counted. For the cis-interactions, we compared the intra-chromosomal interactions (cis-interactions) on the same chromosome of both wild and mutant lines. For example, cis-interactions on chr18 (Table S8) were found to show a great difference between WT (850) and mutant (1275). These results clearly suggest that the number of interacting chromosomes and chromosome length are not necessarily related, because of which, we did not normalized the statistics of interaction.

Comment 5 of the Reviewer#1: Fig5B is not a good example of merge and split. The difference is not as obvious as Fig7. It may be caused by differential mapping due to sequence divergence between WT and mutant instead of true merge and split. In addition, the figure does not indicate what does the upper/lower heat maps and the left/right heatmaps refer to, nor does it explain what the blue and dotted lines between the heat maps refer to.

Authors' Response: We thank the Reviewer for sharing observations on Fig 5B and Fig 7. Based on the suggestions, we have now replaced the Fig 5B with a good example to illustrate TAD merge and split. We added the corresponding indications of the upper/lower heat maps in Fig 5B, and explained the meaning of the blue and dotted lines in the legends.

Comment 6 of the Reviewer#1: The resolution of all figures are too low.

Authors' Response: We revised the format and improved the resolution of all figures.

Comment 7 of the Reviewer#1: All legends need more detailed explanation.

Authors' Response: We have included now more detailed explanation for all the legends.

Comment 8 of the Reviewer#1: More detailed statistics of RNA-seq data need to be demonstrated. The data quality, the mapping result, the comparison result, and the repeatability between replicates.

Authors' Response: Many thanks for these suggestions. All the detailed statistics of RNA-seq data, including the data quality, the mapping result, the comparison result, and the repeatability between replicates, were demonstrated in the supplemental Table S3.

Comment 9 of the Reviewer#1: P3L62-64, recently published Hi-C data in wheat need to be cited.

Authors' Response: Many thanks for your comments. Hi-C data in wheat recently published in Genome Biology was cited in both Introduction (Line 76) and Discussion part (Line 417-419) of the revised manuscript.

Comment 10 of the Reviewer#1: Page6 Line136-138 and Fig2C are to illustrate "Centromeric regions in the dwarf mutant (H1314) showed different genomic compartmentalization from A to B type and vice versa in comparison with the wild type". Are all centromeric regions shown in the figure, or is it the yellow shaded region?

Authors' Response: Many thanks for your kind suggestions. This sentence was revised to "Genomic regions in the dwarf mutant (H1314) showed different compartmentalization from A to B type and vice versa in comparison with the wild type". The yellow shaded region in Fig2C was used to show the uneven distribution of A and B compartments on chr08 of wild type.

Comment 11 of the Reviewer#1: The legends of Fig. 4A and 4B are the same. It seems one is

cis- and the other is trans-.

Authors' Response: Many thanks for pointing out this typo error. We have corrected the corresponding legends of Fig 4A and 4B in the revised manuscript.

Comment 12 of the Reviewer#1: Page8 Line192-204, homoeologous region has more interactions, but the expression difference of homoeologous genes is very small. Is the intensity of homoeologous region interaction and expression difference correlated?

Authors' Response: We greatly appreciate for seeking clarification on this important aspect. Studies in cotton (Wang et al., 2018) revealed that more than 60% of inter-subgenomic interactions occur in subgenome-homologous chromosomes, and the effect of inter-subgenomic interactions on transcriptional regulation of homoeologous genes were explored. The results showed that about 25.9% of gene pairs in *G. hirsutum* and 27.0% in *G. barbadense* with expression bias (two-fold change) have chromatin interactions. Similar to the previous studies, the intensity of homoeologous region interaction was not definitely correlated with expression difference in our study too.

General comment from the Reviewer#2: In the manuscript, Zhang and colleagues applied multi-omics methods, such as Hi-C, RNA-Seq and ATAC-seq, to study the difference between wild type and mutant peanut lines, especially about the 3D genome structure. They found that there are changes in the A/B compartments, TADs and interactions between wild type and mutant peanut lines, and some motifs and differential expressed genes are associated with open chromatin regions and chromatin structure changes. Although such findings are observed at the first time in peanuts, conceptually they have been reported in other species, such as cotton. Before the manuscript is considered for publication in *Genome Biology*, some issues need to be addressed.

Authors' Response: Authors are thankful to the Reveiwer#2 for sharing important suggestions which helped us in improving this MS further.

Comment 1 of the Reviewer#2: For 3D genome analysis, one main issue is the quality of the reference genome assembly. As the authors mentioned in Line 119-120: "This study also identified several sequence scaffolding errors that could not be experimentally validated". Such kind of scaffolding errors will affect the identification of 3D genome structure from Hi-C in the downstream analysis. The authors need to discuss this in the manuscript.

Authors' Response: We thank Reviewer for this important observation and seeking clarification on this important aspect. To provide more clarity, we have added now several sentences to discuss about the 3D genome structure and the scaffolding errors in the revised manuscript in DISCUSSION part. "High throughput next-generation sequencing (NGS) technology has allowed to obtain massive amounts of genomic sequences, but it is not sufficient to produce reference-quality genomes, which seriously hinders our genomic view of many critical biological systems, such as large-scale genome evolution, ployploid and rearranged genomes, and 3D genome organization (Oddes et al., 2018). Ghavi-Helm et al (2019) reported that extensive rearrangements in *Drosophila* genome caused many changes to chromatin topology, disrupting long-range loops, TADs. Recently, a notion that Hi-C measurements can be used to solve outstanding challenges related to genome assembly was proposed, and used to scaffold several major genomes including the frog, goat and barley."

Comment 2 of the Reviewer#2: For allotetraploid genomes, one issue is the homoeologous sequences in the subgenome-homoeologous chromosomes. In Lines 184-204 and in Figure 3D-F, the authors discussed the trans-interactions between homoeologous chromosomes. The homoeologous sequences in the subgenome-homoeologous chromosomes will affect the mapping results and the identification of chromatin interactions. The authors need to discuss the possibility of mis-mapped reads from one homoeologous chromosome to another homoeologous chromosome. Such mis-mapping of the reads from the homoeologous chromosomes could affect the identification of the trans-interactions between homoeologous chromosomes.

Authors' Response: We greatly appreciate for this suggestion. The cultivated peanut is an allotetraploid species ($2n = 4x = 40$, AABB) in *Arachis* with two different genomes, A and B (Bertioli et al. 2019). In view of the high repetitive nature of genomes and large genome size (estimated to be 2.7 Gb) of peanut, Hi-C data was analyzed using Trimmomatic (version 0.38) for quality filtering, then the clean Hi-C data were mapped to the reference genome sequence using the bowtie 2 software (version 2.3.3). For each line, contact maps were constructed with the pipeline HiC-Pro software (version 2.7.1) with default settings. All the interaction frequency was performed in the peanut genome using the packages HiTC v1.26.0 and ggplot2 v3.1.0, which is similar to the previous report in cotton (Wang et al., 2018). This process inevitably resulted in some mis-mappings between homologous subgenomes in allotetraploid peanut.

Comment 3 of the Reviewer#2: In Figure 1, TADs are not clear, although there are marked triangles in the heatmaps.

Authors' Response: We revised the format and improved the resolution of the figures.

Comment 4 of the Reviewer#2: In Figures 1 and 2, it is better to compare the heatmaps from the wild-type and mutant. It is interesting to know whether there are differences in the 3D genome structure between the wild-type and the mutant.

Authors' Response: Many thanks for your kind suggestions. It is a good idea to compare the heatmaps from the wild-type and mutant in Figure 1 or 2. In view that the genome-wide chromatin interaction map was illustrated at the 500 kb resolution, it was hard to explore the differences between two lines. 3D model of the whole chromosomes in the wild type (H2014) and the dwarf mutant (H1314) were shown in Figure 1. The genome-wide chromatin interaction map of the mutant type was shown in supplemental files.

Comment 5 of the Reviewer#2: Figure 2: How to define "different compartments"?

Authors' Response: "Different compartments" in Figure 2 means all the switching compartments, including A to B compartments and B to A compartments.

Comment 6 of the Reviewer#2: In Fig 5B, I cannot see some difference in the heatmap, although some lines are marked near the heatmap.

Authors' Response: Many thanks for pointing this non-clarity in Fig5B. We have now replaced the Fig 5B with a good example to illustrate TAD merge and split. We also added the corresponding indications of the upper/lower heat maps in Fig 5B, and explained the meaning of the blue and dotted lines in the legends.

Comment 7 of the Reviewer#2: The "conclusions" in the abstract is more like a part of the results, not a real conclusion.

Authors' Response: We have revised "conclusions" in the abstract.

Comment 8 of the Reviewer#2: The bin size in TAD analysis is 50 kb, which is quite big for motif analysis (lines 241-244). Need some detailed explanation for the motif analysis here.

Authors' Response: Thanks for identifying this typo error. Due to the large genome size, we chose 40 kb as a bin to analyze the TAD domain. Motif calling was analyzed on the whole genome using the MEME software, and all motifs were filtered with $q\text{-value} < 0.0001$ and $q\text{-value} < 0.001$. We identified the TAD boundaries by calculating the insulation plot of the 40 kb resolution genome-wide interaction maps, and named each bin on both side of one TAD as a border for calculating the enrichment of motifs. So the motif analysis was also consistent with the same bin size.

Comment 9 of the Reviewer#2: For Figure 3B, the legend is "(B) Cis-interactions identified in wild type and dwarf mutant lines". Figure 3B is more like trans-interactions in one peanut line, either wild type of mutant line.

Authors' Response: Many thanks for identifying this typo error which we have corrected now. Yes...it shows the trans-interactions in the wild type H2014.

Comment 10 of the Reviewer#2: In Fig 4, the figure legend is as follows:

"(A) Venn diagram for all cis-interactions identified in the two peanut lines.

(B) Venn diagram for all cis-interactions identified in these two lines."

Is there something wrong in these two legends? (B) should be about trans-interactions, right?

Authors' Response: Again we apologize for this typo error. We revised the corresponding legends of Fig 4A and 4B in the revised manuscript.

Comment 11 of the Reviewer#2: Figure 7C is blur.

Authors' Response: Many thanks for your comments. Now we have changed the format of Figure 7 and improved the resolution for better exhibition in the revised manuscript.

Comment 12 of the Reviewer#2: There are inconsistencies between the TAD resolution in the method part (lines 494-498) and the TAD resolution in the main text (line 242). Need to clarify it.

Authors' Response: Many thanks for your comments. The TAD resolution was 40 kb. We revised our writing errors in the method part (Line 593-605).

Hope the revised version of manuscript is acceptable now for publication in Genome Biology.

Second round of review

Reviewer 1

The authors addressed most of my concerns. There is still one issue need to be carefully addressed. There are multiple errors in Figure 7, which displayed the most important result.

- 1) The track labels in A and B are exactly the same (H2014)
- 2) Fig. 7B, the TAD indicated by the third lane is not consistent with the heatmap shown on the top
- 3) the gene and the putative enhancer need to be marked on the snapshot.
- 3) The insulator score track is truncated.
- 4) The fifth line is not the ATAC peak, but rather the difference. There is no indication of where the ATAC peaks are.

Reviewer 2

The authors have made some revisions to their manuscript. However, they have not responded to my questions properly. The comments to their responses are as follows.

For the original Comment 1, "scaffolding errors will affect the identification of 3D genome structure from Hi-C in the downstream analysis". The authors just cited the similar questions raised by other researchers. What they need to do is to assess the scaffolding errors in the peanut genome sequences and the effect on their analysis results.

For the original Comment 2, "The authors need to discuss the possibility of mis-mapped reads from one homoeologous chromosome to another homoeologous chromosome". The authors did not assess the mis-mapping in their data. They just rephrased the data analysis process in their study.

For the original Comment 3, the authors updated the figure. However, the TADs are not clear from the updated figure. If this is a better figure for TAD in their data, it should be hard to observe the clear TADs in other regions.

For the original Comment 6: The authors updated Fig 5B. However, the new figure has the same problem as the previous one. I still cannot see TAD merge and split between the wild-type and the mutant.

For the original Comment 8: The authors replied that they used 40 kb for TAD analysis. Still, 40 kb is big for motif calling. They didnot answer the question how such a big resolution was used for motif analysis.

Authors Response

Point-by-point responses to the reviewers' comments:

Authors' response for GBIO-D-20-01475R2

Title: Chromatin Spatial Organization of Wild Type and Mutant Peanuts Reveals High-Resolution Genomic Architecture and Interaction Alterations

Message from the Associate Editor: Thank you very much for submitting your manuscript entitled to Genome Biology, and please accept my apologies for the delay in replying to you about it. It has now been seen by the two original referees and their comments are accessible below.

You will see that while Referee 1 is now mostly satisfied with the manuscript, Referee 2 is still concerned about how scaffolding errors and mis-mapping may be affecting downstream analyses. I am afraid this outstanding issue is substantive and we must insist that it is addressed in the form of a further, final revision of the manuscript, before we reach a final decision on publication. Please note that we are willing to consult the referees on your manuscript only once more, and we will only publish your manuscript if you can properly address all the concerns and persuade Reviewer 2 that these potential errors are not influencing analyses and conclusions. If you do decide to revise your manuscript one last time for Genome Biology, we would hope to receive your further revised manuscript, together with a list of the changes made (specifying where in the manuscript the changes have been made and, if possible, submitting a copy of this manuscript as an extra Additional Data File, highlighting these changes in the text), within the next four weeks or so. Please let us know if the delay is likely to be longer than four weeks, or if you have any problems or questions.

Authors' Response: Authors are thankful to the Handling Associate Editor for arranging the feedback and comments from two Reviewers. We are pleased to note that one Reviewer is satisfied with our revision while second Reviewer wants some more clarification. We also thank you for providing us opportunity to further revise this MS. We have further revised the MS with suggested inputs second Reviewer and provided responses to all the comments. We have submitted the revised files and we are hopeful that Reviewers and you will find the revised version acceptable for publication in Genome Biology.

Comment 1 of the Reviewer#1: The authors addressed most of my concerns. There is still one issue need to be carefully addressed. There are multiple errors in Figure 7, which displayed the most important result. 1) The track labels in A and B are exactly the same (H2014); 2) Fig. 7B, the TAD indicated by the third lane is not consistent with the heatmap shown on the top; 3) the gene and the putative enhancer need to be marked on the snapshot; 3) The insulator score track is truncated; 4) The fifth line is not the ATAC peak, but rather the difference. There is no indication of where the ATAC peaks are.

Authors' Response: We are glad to note that Reviewer#1 is happy with the revision. Also, we thank the Reviewer#1 for indicating these errors in Fig 7 which we left somehow during finalization. We have now corrected above suggested edits in revised Figure 7.

Comment 1 of the Reviewer#2: The authors have made some revisions to their manuscript. However, they have not responded to my questions properly. The comments to their responses are as follows.

For the original Comment 1, "scaffolding errors will affect the identification of 3D genome structure from Hi-C in the downstream analysis". The authors just cited the similar questions raised by other researchers. What they need to do is to assess the scaffolding errors in the peanut genome sequences and the effect on their analysis results.

Authors' Response: We appreciate Reviewer#2 for suggestions on analysis regarding detection of scaffolding errors. We performed analysis using Hi-C data and identified several scaffolding errors by the whole-genome interaction map, such as inversions on chr5 and chr11... (List below) and these errors would influence the number of TADs and A/B compartments. In our study, we mainly focused on significantly different TADs between the wide type and the mutant. In order to assess the main results, we further conducted the co-linearity analysis among the three reference genomes, including Tifrunner, *A. monticola*, and Shitouqi. The co-linearity results showed the high quality of genome assembly of chr19, which cannot influence the main results in this study (see following figures). We also performed 3C assay to validate the key chromatin loop on chr19 (interaction between 23,040,000 and 23,160,000). The qPCR results provided evidence for the chromatin interaction (Supplementary Fig.7 and Table S18). Keeping in mind above results, we have revised the corresponding sentences in the revised manuscript (Line 414-423).

List of major scaffolding errors:

- chr2 36.8M-45.12M **Inversion**
- chr3 52.24M-96.36M **Rearrangement**
- chr5 61.54M-64.92M **Inversion**
- chr11 23.88M-34.48M **Inversion**
- chr14 24.72M-34.12M **Inversion**
- chr14 77.24M-90.4M **Translocation**
- chr15 51.68M-81.24M **Inversion**
- chr16 59.44M-61.48M **Inversion**
- chr20 80.24M-88.68M **Inversion**
- chr3:133M-143M chr13:135M-146M **Interchromosomal translocation**
- chr6:107M-113M chr16:143M-152M **Interchromosomal translocation**

The co-linearity analysis among the three reference genomes

Comment 2 of the Reviewer#2: For the original Comment 2, "The authors need to discuss the possibility of mis-mapped reads from one homoeologous chromosome to another homoeologous chromosome". The authors did not assess the mis-mapping in their data. They just rephrased the data analysis process in their study.

Authors' Response: We appreciate Reviewer#2 for raising the issue of mis-mapped reads between homoeologous chromosomes. In order to assess the mis-mapped reads from one homoeologous chromosome to another, we used MUMMER software for sequence alignment in the whole-genome level (chr1-chr20) and the homoeologous sub-genome level (chr1-chr11, chr2-chr12, ...). Results showed that the sequence

similarity between homoeologous sub-genome were low (see following figures). Then we used Juicer software to analyze sequence alignment (IGV), and found that only a few INDELs and heterozygous SNPs were identified between homoeologous sub-genome.

In order to provide convincing evidences, we further extracted the reads from the bam files, especially mapped to homoeologous sub-genomes. For example, all the reads mapped to chr1 and chr11 were extracted, respectively. Then the reads mapped to chr1 were used for cross-comparison with those reads from chr11 (alignment parameters were set as: `-X 1000 --no-unal --very-sensitive-local --no-mixed --no-discordant`). Results showed that the sequence alignment was very low in view of the plenty of mismatch. All these results showed that the possibility of mis-mapped reads from one homoeologous chromosome to another homoeologous chromosome was very low in Hi-C analysis.

Chr. 1 vs Chr. 11

genome_compare

IGV_chr1_18M

IGV_chr11_18M

Chr1 alignment on chr11

Chr11 alignment on chr1

Comment 3 of the Reviewer#2: For the original Comment 3, the authors updated the figure. However, the TADs are not clear from the updated figure. If this is a better figure for TAD in their data, it should be hard to observe the clear TADs in other regions.

Authors' Response: We thank the Reviewer#2 for this comment. We further conducted the TAD aggregate plot, and the results showed that the border of TADs was obvious (see following figures). Not all the TADs identified in plants were clear as those in mammalian. One possible reason might be due to the tissue used for TAD analysis consist of different cell types with heterogeneity, which was totally different from the pure cell line used in human. Here, we added the insulation curves for the better exhibition of TADs identified in chr4:75Mb~87Mb and chr7:38Mb~50Mb.

We also tried different methods to identified TADS, such as the Insulation of TAD-tool software. We took chr4 for example, and found that the TADs identified by the Insulation values were almost consistent with those mentioned in the manuscript (see following figures).

H2014_TAD aggregate plot

H1314_TAD aggregate plot

Insulation C-word in this study

Insulation TAD-tool

Comment 4 of the Reviewer#2: For the original Comment 6: The authors updated Fig 5B. However, the new figure has the same problem as the previous one. I still cannot see TAD merge and split between the wild-type and the mutant.

Authors' Response: Based on the suggestion, we have revised Fig 5B in the revised manuscript. In order to show clear TAD merge and split between the wild-type and the mutant, we added the insulation scores.

Comment 5 of the Reviewer#2: For the original Comment 8: The authors replied that they used 40 kb for TAD analysis. Still, 40 kb is big for motif calling. They did not answer the question how such a big resolution was used for motif analysis.

Authors' Response: We completely agree with Reviewer#2 that 40 kb resolution was a little big for motif calling. In this study, the length of the border of TAD were 40kb, and the total length were 133.72 Mb in H1314 and 133.32 Mb in H2014, respectively. We first scanned the potential TF binding site genome-wide followed by checking the overlapping sequences in the TAD borders, and finally searching the overlapping sites between TAD borders and TF binding sites. Here, we mainly focused on the difference of TADs and motifs between the wild-type and the mutant. Both the

top five motifs overlapping the TAD borders and inside of borders in the wide type and the mutant were listed below.

The top five motifs overlapping the TAD borders in H2014 (left) and H1314 (right)

Motif ID	Motif name	P-value	Motif ID	Motif name	P-value
MA0044.1	HMG-1	0.6444046	MA0044.1	HMG-1	0.3379101
MA1199.1	AGL16	0.01814938	MA1206.1	ARF2	0.9995484
MA1273.1	dof4.2	0	MA0555.1	SVP	0.2640525
MA1375.1	ANL2	0.9997426	MA1375.1	ANL2	0.9999997
MA1394.1	MYB73	0.006573196	MA1394.1	MYB73	0.0001973989

The top five motifs enriched in the TAD borders in H2014 (left) and H1314 (right)

Motif ID	Motif name	P-value	Motif ID	Motif name	P-value
MA0590.1	LFY	0	MA0551.1	HY5	0
MA1273.1	dof4.2	0	MA0590.1	LFY	0
MA1380.1	AT2G20110	0	MA1380.1	AT2G20110	0
MA1367.1	AT1G76870	8.88E-16	MA1366.1	AT1G76880	8.22E-15
MA0551.1	HY5	1.54E-14	MA1045.1	NAC043	3.16E-13

Third round of review

Reviewer 2

The authors have addressed my concerns in the revised manuscript. I have no further questions.